# Continuous Subcutaneous Insulin Infusion (CSII) Combined with Oral Glucose-Lowering Drugs in Type 2 Diabetes: A Systematic Review and Network Meta-Analysis of Randomized, Controlled Trials

**DOI:** 10.3390/ph15080953

**Published:** 2022-07-30

**Authors:** Hui Li, Aimin Yang, Shi Zhao, Elaine YK Chow, Mohammad Javanbakht, Yinhui Li, Dandan Lin, Lijuan Xu, Deng Zang, Kai Wang, Li Ma

**Affiliations:** 1Department of endocrine, Traditional Chinese Medicine Hospital Affiliated to Xinjiang Medical University, Urumqi 830000, China; 407453717hui@gmail.com (H.L.); liyinhui0513@gmail.com (Y.L.); xulijuan645@gmail.com (L.X.); dengzang629@gmail.com (D.Z.); 2Department of Medicine and Therapeutics, Chinese University of Hong Kong, Hong Kong 999077, China; aiminyang@cuhk.edu.hk (A.Y.); e.chow@cuhk.edu.hk (E.Y.C.); 3JC School of Public Health and Primary Care, Chinese University of Hong Kong, Hong Kong 999077, China; shi.zhao@link.cuhk.edu.hk; 4Nephrology and Urology Research Center, Baqiyatallah University of Medical Sciences, Tehran 1435916471, Iran; mhmjvbt81@gmail.com; 5Department of Medical Engineering and Technology, Xinjiang Medical University, Urumqi 830011, China; lindandan199113@gmail.com

**Keywords:** type 2 diabetes, insulin infusion (CSII), oral hypoglycemic drugs, RCT, network meta-analysis

## Abstract

The clinical efficacy of continuous subcutaneous insulin infusion (CSII) therapy combined with six classes of oral glucose-lowering drugs (GLDs) (TZDs/metformin/acarbose/GLP-1 receptor agonist/SGLT-2 inhibitor/DPP-4 inhibitor) was evaluated by a network meta-analysis to provide an evidence-based reference in making a clinical decision on CSII combined with drugs in the treatment of type 2 diabetes. Data were retrieved from eight databases: the Chinese Journal Full-Text Database (CNKI), VIP Chinese Science and Technology Periodicals Full-Text Database (VP-CSFD), Wanfang Data Journal Paper Resource (WANFANG), China Biomedical Database (CBM), PubMed, Embase, Cochrane Library, and Web of Science. The retrieval period dated from the library’s construction to 27 June 2021. The search was for randomized, controlled trial studies (RCT) on insulin infusion (CSII) combined with oral hypoglycemic drugs (TZDs/metformin/acarbose/GLP-1 receptor agonist/SGLT-2 inhibitor/DPP-4 inhibitor) in the treatment of type 2 diabetes. Quality evaluation and data extraction were performed on the studies included, and network meta-analysis was performed with R4.0.1 software. A total of 56 publications was included in the final network meta-analysis, with a total sample size of 4395. Results based on the network meta-analysis were that CSII combined with a metformin works best on fasting blood glucose (FBG) and 2 h postprandial blood glucose (2hPG) and improves insulin resistance (lower HOMA-IR levels). CSII combined with a DPP-4 inhibitor had the best clinical effect in reducing glycosylated hemoglobin levels. Treatment with CSII combined with a DPP-4 inhibitor was the fastest way to achieve the blood glucose standard. In terms of insulin dosage, an insulin pump (CSII) combined with the GLP-1 receptor agonist can significantly reduce insulin dosage. Network meta-analysis evidence suggests that an insulin infusion (CSII) combined with oral hypoglycemic drugs can improve clinical efficacy in controlling blood sugar and improving insulin resistance, insulin dosage, and standard time. However, the most outstanding performance was that of insulin infusion (CSII) combined with metformin, which had the best clinical effect in controlling blood sugar and improving insulin resistance.

## 1. Introduction

In 2021, half a billion people or 10.5% of the world’s population were affected by diabetes, the majority having type 2 diabetes (T2D) with the progressive nature characterized by elevated blood glucose [1]. Pharmacological treatment plays an important role in the treatment of T2D. Many individuals with T2D eventually require and benefit from insulin therapy [2]. Continuous subcutaneous insulin infusion (CSII), an alternative to multiple daily injections (MDI) in intensive insulin therapy for optimizing glycemic control in T2D, was associated with improved long-term outcomes [3,4]. CSII via pump therapy had modest advantages for lowering HbA1c and for reducing severe hypoglycemia rates in adults [5]. Compared with the widespread use of CSII in type 1 diabetes, little evidence is available on the effects of combination CSII therapy for the treatment of T2D [6].

The combination of insulin therapy and oral glucose-lowering drugs (GLDs) is a potential approach in T2D patients with severe insulin resistance and poor glycemic control. For example, the combination of basal insulin and glucagon-like peptide 1 receptor agonist (GLP-1 RA) has potent glucose-lowering actions and less weight gain and hypoglycemia compared with intensified insulin regimens [7]. In the latest international guideline, combination insulin therapy with GLP-1 RA is also recommended for a greater efficacy and durability of the treatment effect [2]. CSII is an important and effective method of insulin pump intensive treatment. Intensive therapy with multiple drugs in the early stage of T2D could also correct multiple pathophysiological defects, improve clinical efficacy, improve insulin resistance, and protect β-cell function [8]. The glycemic efficacy of the combination of insulin therapy with oral GLDs varied in previous studies including in controlling glucose, improving insulin resistance, increasing insulin sensitivity, accelerating the time of blood glucose reaching the target, and reducing insulin dosage [9,10]. However, a comparison of efficacy in insulin therapy, especially CSII via pump therapy, combined with oral GLDs in T2D was not addressed in previous studies that used a conventional meta-analysis on a pair-based comparison. Network meta-analysis is an extension of conventional meta-analysis with the advantage of evaluating and comparing the efficacy of multiple interventions [11].

In this study, we conducted a systematic review and network meta-analysis of randomized, controlled trials (RCT) to evaluate the clinical efficacy of the combination of CSII with six classes of oral GLDs for clinical decision making on the treatment of T2D.

## 2. Materials and Methods

We registered this study on PROSPERO (CRD42022307702) and followed the Preferred Reporting Items for Systematic Reviews and Meta-Analyses guideline (PRISMA-2020) [12] and the extension statement for network meta-analysis (PRISM-NMA) [13].

### 2.1. Inclusion and Exclusion Criteria

Eligible RCTs included (1) patients with T2D regardless of the course of the disease and the case origin and (2) RCTs that compared simple CSII and CSII combined with six classes of GLDs (thiazolidinediones, metformin, AGIs, GLP-1 RAs, SGLT-2is, DPP-4is). (The final analysis did not specify a detailed list of specific drugs in each class of GLD. Due to the increased risk of hypoglycemia associated with the combined use of sulfonylureas and insulin, there have been few such RCTS; therefore, sulfonylureas and CSII were not included here.); and (3) those that reported the following outcomes: blood glucose indexes before and after treatment (fasting plasma glucose, 2 h post-meal blood glucose, HbA1c), islet function (fasting C-peptide, 2 h post-meal C-peptide, β-cell function index (HOMA-β), insulin resistance index (HOMA-IR), blood glucose standard time, and insulin dosage. We excluded RCTs for these reasons: (1) patients with T2D and with other diseases; (2) research design was not clearly defined; (3) outcomes were not available; (4) duplicate literature; (5) full text of the literature was not available; and (6) CVD, hypertension, cancer, CKD, and other T2D patients.

### 2.2. Retrieval Strategy

We searched eight databases including four English databases (PubMed, Embase (Netherlands), Cochrane Library (Britain), and Web of Science (United States)) and four Chinese databases (CNKI, VP-CSFD, WANFANG, and CBM) for RCTs from inception to 27 June 2021. The databases were searched by a combination of subject and free search, and the RCTS followed the search strategy (Appendix A).

### 2.3. Paper Screening, Data Extraction, and Quality Evaluation

Two researchers independently completed the literature screening by reading the title and abstract in NoteExpress (version 2.5.1.1154, Beijing Aegean Software Company, Beijing, China). Discrepancies were resolved by a third adjudication. We extracted the following information using a pre-designed collection form: study characteristics (authors, publication year, study location, and design); interventions (number of intervention and control groups, intervention measures); outcome measures (FPG, 2 h PG, HbA1C, HOMA-IR, time of blood glucose standards, and insulin consumption); and research bias indicators and quality assessment.

The Cochrane collaborative risk assessment tool was used to assess the quality of each study [14]. The quality evaluation was carried out independently by two researchers and cross-checked. In the case of any disagreement, this was resolved through discussion and/or consulting third-party opinions.

### 2.4. Statistical Analysis

We used the Bayesian analytical framework to perform random effects network meta-analysis and to calculate the mean differences (MD) in the outcomes for continuous measurement. When the mean and standard deviation of the difference were not available, we used the mean and standard deviation of the measured value after the intervention in the original literature for analysis [15].

The posterior distribution of MD was inferred to quantify the curative effect of different interventions. The plot of a network visually expresses the evidence relationship and concisely describes its characteristics, which show both the overall status of the network and the numbers of studies and responses to treatments [16]. We used the surface under the cumulative ranking curve (SUCRA) value and ranking probability plot to detect and explain the efficacy of each intervention. We used the node-splitting method to compare direct and network evidence, and we estimated the inconsistency between direct and network comparisons [17,18]. We used posterior distribution of τ^2^ and I^2^ statistics to assess the heterogeneity of treatment effects among different studies.

The network meta-analysis based on the Bayesian framework was performed by using the package “*gemtc*” (version 1.0-0) in R (version 4.0.1) software [19]. The package “*rjags*” (version 4-10) was adopted to implement Just Another Gibbs Sampler (JAGS, version 4.3.0) to perform Markov chain Monte Carlo (MCMC) operations [20].

## 3. Results

### 3.1. Literature Retrieval

Of 697 publications identified (599 Chinese and 98 English), we assessed 56 publications in the final network analyses of 4183 participants with T2D (Figure 1 and Appendix A).

### 3.2. General Information of the Papers

The risk of bias was high in 52 studies and unclear in 4 studies (Figure 2 and Figure 3). The details in quality assessment are summarized in Appendix A. Figure 4 shows the network plots for each glycemic outcome by different comparisons. Except for CSII combined with SGLT2is, other interventions included outcome indicators for FPG, 2h-PG, and HBA1C. Except for CSII combined with acarbose, other interventions included outcome indicators for HOMA-IR. All interventions included outcome indicators for insulin dosage. Except for CSII combined with GLP-1 receptor agonists, other interventions included outcome indicators for time for the blood sugar to reach standards.

### 3.3. Network Meta-Analysis Results

#### 3.3.1. FPG

As can be seen in Figure 5, compared with the CSII group (A), in groups C, G, and B, the fasting blood glucose level was significantly reduced. The group CSII combined with metformin had the largest reduction in fasting blood glucose (Group C) (MD = −1.2, 95% CI (−1.7, −0.81)) followed by CSII combined with DPP-4 inhibitors (Group G) (MD = −0.76, 95% CI (−1.1, −0.37)) and CSII combined with TZDs (Group B) (MD = −0.6, 95% CI (−0.97, −0.23)). SUCRA was used to summarize the six treatments and rank them in terms of efficacy of fasting glucose. The results shown in Table 1 and Figure 6 suggest that the decrease in the degree of fasting blood glucose was ranked as follows: CSII + Metformin/MF (C) > CSII + DPP-4is (G) > CSII + AGIs (D) > CSII + TZDs (B) > CSII + GLP-1RAs (E) > CSII (A). This indicates that loading metformin results in the largest reduction in fasting glucose. The specific results of the comparison of the effects of insulin infusion (CSII) combined with six different types of oral hypoglycemic drugs on the decrease in the degree of fasting blood glucose are shown in the Appendix A.

#### 3.3.2. The 2 h PG

As can be seen in Figure 5, compared with the CSII group (A), in Groups C, G, and B, the 2 h PG level was significantly reduced. In the group in which CSII was combined with metformin, there was the largest reduction in 2 h PG (MD = −2.1, 95% CI (−2.8, −1.4)) followed by CSII combined with DPP-4 inhibitors (Group G) (MD = −1.4, 95% CI (−1.9, −0.85)) and CSII combined with TZDs (Group B) (MD = −0.61, 95% CI (−1.2, −0.026)). SUCRA was used to summarize the six treatments and rank them in terms of efficacy of 2 h PG. Table 1 and Figure 6 show the results for the reduction in 2 h PG, CSII + Metformin/MF (C) > CSII + DPP-4is (G) > CSII + GLP-1RAs (E) > CSII + TZDs(B) > CSII + AGIs (D) > CSII(A). This indicates that loading metformin results in the largest reduction in 2 h PG. The specific results of the comparison of the effects of insulin infusion (CSII) combined with six different types of oral hypoglycemic drugs on the 2 h PG level are shown in the Appendix A.

#### 3.3.3. HbA1c

As can be seen in Figure 5, compared with the CSII group, in Groups G (CSII + DPP-4is), E (CSII + GLP-1Ras), and B (CSII + TZDs), the HbA1C level was significantly reduced. In the group in which CSII was combined with the DPP-4is (Group G), there was the largest reduction in HbA1C (MD = −0.87, 95% CI (−1.2, −0.55)) followed by the CSII + GLP-1RAs (Group E) (MD = −0.59, 95%CI (−1.1, −0.078)) and CSII + TZDs (Group B) (MD = −0.37, 95% CI (−0.67, −0.074)). SUCRA was used to summarize the six treatments and rank them in terms of the efficacy of HbA1C. As can be seen in Table 1 and Figure 6, the ranking for the reduction in HbA1C was as follows: CSII + DPP-4is (G) > CSII + GLP-1RAs (E) > CSII + TZDs (B) > CSII + Metformin/MF (C) > CSII + AGIs (D) > CSII (A). This indicates that loading DPP-4 inhibitor results in the largest reduction in HbA1C. The specific results of the comparison of the effects of insulin infusion (CSII) combined with the six different types of oral hypoglycemic drugs on the HbA1C level are shown in the Appendix A.

#### 3.3.4. HOMA-IR

As can be seen in Figure 5, compared with the CSII group (A), in Groups C, G, and B the HOMA-IR level was significantly reduced. The group in which CSII was combined with metformin showed the largest reduction in HOMA-IR (MD = −0.59, 95% CI (−0.92, −0.26)) followed by CSII + DPP-4is (Group G) (MD = −0.45, 95% CI (−0.74, −0.17)) and CSII + TZDs (Group B) (MD = −0.32, 95%CI (−0.55, −0.097)). SUCRA was used to summarize the six treatments and rank them in terms of efficacy in reducing HOMA-IR. As can be seen in Table 1 and Figure 6, ranking for the reduction in HOMA-IR was as follows: CSII + Metformin/MF (C) > CSII + DPP-4is (G) > CSII + TZDs (B) > CSII + GLP-1RAs (E) > CSII (A) > CSII + SGLT2is (F). This indicates that loading metformin results in the largest reduction in HOMA-IR. The specific results of the comparison of the effects of insulin infusion (CSII) combined with six different types of oral hypoglycemic drugs on the HOMA-IR level are shown in the Appendix A.

#### 3.3.5. Insulin Dosage when Blood Glucose Reaches Standard (Insulin Dosage)

As can be seen in Figure 5, compared with CSII group (A), less insulin was required in all the other groups to meet their blood glucose standard. In the group CSII+GLP-1RAs (E), the minimum amount of insulin was required (MD = −15, 95% CI (−21, −9.8)) followed by the group CSII + SGLT2is (Group F) (MD = −11, 95% CI (−17, −5.7)). SUCRA was used to summarize the seven treatments and rank them in terms of insulin dosage. As can be seen in Table 1 and Figure 6, ranking was from low to high according to the amount of insulin used: CSII + GLP-1RAs (E) > CSII + SGLT2is (F) > CSII + AGIs (D) > CSII + Metformin/MF (C) > CSII + DPP-4is (G) > CSII + TZDs (B) > CSII (A). This indicates that the minimum amount of insulin is required to load the GLP-1 receptor agonist. The specific results of the comparison of the effects of insulin infusion (CSII) combined with six different types of oral hypoglycemic drugs on the change in insulin dosage are shown in the Appendix A.

#### 3.3.6. Time for Blood Sugar to Reach Standard

As can be seen from Figure 5, compared with the CSII group (A), except for Group D, in the other groups, blood glucose standards were reached within a short time. The group CSII + DPP-4is (G) had the fastest blood sugar standard time (MD = −3.4, 95% CI (−4.2, −2.5) followed by the group CSII + SGLT2is (Group F) (MD = −2.9, 95% CI (−4.9, −1.0)). SUCRA was used to summarize the six treatments and rank them in terms of blood glucose attainment time. Table 1 and Figure 6 show the time taken for blood glucose to reach standard, from short to long, as follows: CSII + DPP-4is (G) > CSII + SGLT2is (F) > CSII + AGIs (D) > CSII + Metformin/MF (C) > CSII + TZDs (B) > CSII (A). This indicates that the time it takes for blood glucose to reach standard was shortest when the dPP-4 inhibitor was loaded. The specific results of the comparison of the effects of insulin infusion (CSII) combined with six different types of oral hypoglycemic drugs on the change in the time for blood sugar to reach standard are shown in the Appendix A.

### 3.4. Network Heterogeneity and Inconsistency

We compared the DIC of the consistency model with the DIC of the inconsistency model. If the difference was within 5, it indicated that the data basically met the premise of consistency and follow-up research could be carried out. We established a fixed effect model and random effect model for all outcome indicators. The results showed that the I^2^ of the fixed effect model was greater than 50%, but the I^2^ of the random effect model was less than 1%. At the same time, for all outcome indicators, the DIC and mean of the posteriori residuals of the random effect model were less than those of the fixed effect model (Appendix A). Therefore, the random effect model was used for meta-analysis. The global heterogeneity assessment found that I^2^ > 70% for all the different outcome indicators, indicating a high global heterogeneity (Appendix A). Inconsistency between direct and indirect comparisons of different outcome indicators was assessed by a node-splitting approach, comparing estimates from direct and indirect evidence. The Appendix A show that the *p* values for the inconsistency test for all the outcome indicators were greater than 0.05 and there was no local inconsistency (Appendix A). The trace map, density map, and convergence diagnosis diagram for all outcome indicators are shown in the Appendix A.

## 4. Discussion

A total of 56 publications were included in this study, with a total sample size of 4183 patients. The study used Bayesian network meta-analysis to compare the clinical efficacy of the use of an insulin pump combined with six oral hypoglycemic drugs (TZDs/metformin/acarbose/GLP-1 receptor agonist/SGLT-2 inhibitor/DPP-4 inhibitor). Indexes reflecting blood glucose levels were selected, including fasting blood glucose, 2 h postprandial blood glucose, and glycosylated hemoglobin. The network meta-analysis selected the following indicators: (1) there were the following clinical indicators of blood glucose levels: FPG, 2hPG, and HbA1C; (2) the following indicators reflected insulin sensitivity, insulin resistance levels, and islet β-cell function: fasting C peptide, 2 h postprandial C peptide, fasting insulin, 2 h postprandial insulin, HOMA-IR, and HOMA-β; (3) the following indicators reflected the clinical effects: the time blood glucose took to reach the standard and the amount of insulin used when blood glucose reached the standard; and (4) the following indicator reflected the safety of treatment: the hypoglycemic incidence. In the final mesh meta-analysis, the numbers for fasting C-peptide, 2 h postprandial C-peptide, fasting insulin, 2 h postprandial insulin, HOMA-β, and the incidence of hypoglycemia were small, which did not meet the data requirements of the mesh meta-analysis. Therefore, the above six indicators were not analyzed in this paper. Therefore, six indicators were finally analyzed. The six indicators were FPG, 2hPG, HbA1C, HOMA-IR, blood glucose standard time, and insulin dosage when blood glucose standard was reached.

In terms of the control of blood sugar and reduced insulin resistance, an insulin pump (CSII) combined with biguanide, DPP-4 inhibitors, sensitizers, and GLP-1 receptor agonists can further reduce FPG, 2hPG, and HOMA-IR levels compared with an insulin pump alone [9,21,22,23,24,25]. Among them, the combination of biguanide had the best efficacy, followed by DPP-4 inhibitors. At present, a systematic evaluation of CSII combined with oral hypoglycemic drugs is still lacking, but a number of previous randomized, controlled trials has suggested that combined oral hypoglycemic drugs have a better hypoglycemic effect than insulin infusion therapy. This study also further confirms the clinical advantages of metformin and other oral hypoglycemic drugs used with an insulin pump. Metformin can improve sugar metabolism and inhibit liver glycogen by enhancing the absorption and utilization of glucose in peripheral tissues, so as to achieve the effect of lowering glucose and increasing insulin sensitivity. According to clinical practice, metformin combined with insulin is relatively safe [26]. It can reduce blood glucose and improve hyperinsulinemia and insulin resistance with a good clinical effect. In terms of the control of HbA1C, the combination of CSII therapy with TZDs, metformin, acarbose, GLP-1 receptor agonist, and DPP-4 inhibitor reduces the degree of HbA1C more than that of insulin infusion (CSII) alone; in combination with DPP-4 inhibitor, the degree of HbA1C is most reduced, followed by the combination of GLP-1 receptor agonist. DPP-4 inhibitors can inhibit glucagon secretion by increasing endogenous GLP-1 in patients and can be used in combination with insulin to improve insulin resistance, eliminate hyperglycemic toxicity altogether, complement the mechanism, and control glucose for a long time. In terms of insulin dosage [10,26,27,28], CSII therapy combined with TZDs, metformin, acarbose, GLP-1 receptor agonist, SGLT-2 inhibitor, and DPP-4 inhibitor uses less insulin than CSII therapy alone. The GLP-1 receptor agonist has the best effect, followed by a combination with the SGLT-2 inhibitor. CSII therapy, combined with metformin, acarbose, SGLT-2 inhibitor, and DPP-4 inhibitor, can accelerate the time it takes the blood glucose to reach a standard. Of these combinations, the combination with the DPP-4 inhibitor is the fastest, followed by the combination with the SGLT-2 inhibitor. The contribution of GLP-1 receptor agonists to blood glucose attainment time was again not discussed, due to the failure to include the relevant RCT literature on GLP-1 receptor agonists. GLP-1RA regulates blood glucose through a variety of mechanisms: increasing islet β-cell proliferation, decreasing islet β-cell apoptosis, increasing insulin sensitivity, decreasing glucagon secretion, enhancing glucose utilization in peripheral tissues, and decreasing hepatic gluconeogenesis. The combination of GLP-1RA and insulin has synergistic and complementary effects, which provides a theoretical basis for a combination therapy and can significantly reduce the dosage of insulin. Therefore, insulin infusion (CSII) therapy combined with a GLP-1RA/DPP-4 inhibitor has certain advantages in terms of insulin dosage and blood glucose compliance time.

This study still has some limitations. The quality of the literature as a whole was not high. Most of the studies did not mention the allocation of concealment and blinding. The resulting risk of bias may have some impact on the reliability of the results. The duration of treatment and follow-up in the included studies was different, and there could be the possibility of selection bias, implementation bias, and measurement bias, thus limiting the strength of the evidence obtained in this meta-analysis. There are several areas of concern in a future systematic evaluation, including assessing conflicts of interest, performing a comprehensive search of the literature including gray literature, provision of a detailed list of the included and excluded literature, and plans for systematic evaluation [11]. Due to the small number of studies on HOMA-β, fasting C-peptide, 2 h postprandial C-peptide, and the incidence of hypoglycemia in the literature included, more indicators were not discussed in this paper. Moreover, there was great heterogeneity among the studies due to the differences in treatment course, disease course, and grouping scheme, which affected the reliability of the results. The network meta-analysis method is still at the stage of exploration and learning, and it is possible to cause a certain bias due to the failure to analyze each study individually.

Based on the network meta-analysis of this study, insulin infusion combined with oral hypoglycemic drugs is superior to insulin infusion therapy alone. Moreover, the combination with metformin therapy has the best effect on blood glucose control and insulin resistance reduction. The combination with the GLP-1RA/DPP-4 inhibitor has clinical advantages in that it accelerates the time taken to achieve blood glucose compliance and reduces the dosage of insulin. However, as most of the original studies included in this study were Chinese publications, the inconsistent quality of methodology has a certain influence on the research conclusions. Therefore, the sequencing results should be treated with caution and need to be validated by high-quality RCTs to provide evidence-based support for the use of an insulin pump combined with oral drugs in the treatment of type 2 diabetes.

## Figures and Tables

**Figure 1 pharmaceuticals-15-00953-f001:**
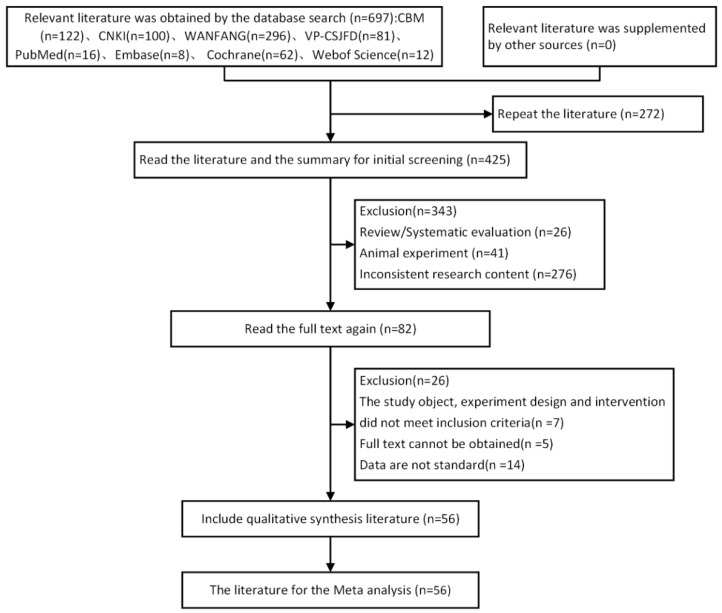
Flowchart of literature screening.

**Figure 2 pharmaceuticals-15-00953-f002:**
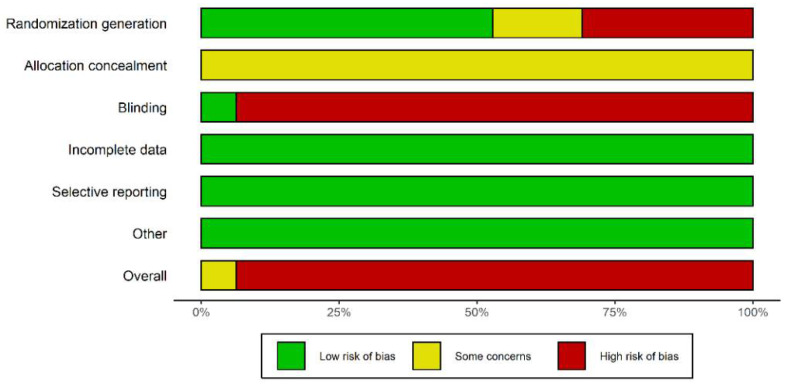
Risk of bias graph.

**Figure 3 pharmaceuticals-15-00953-f003:**
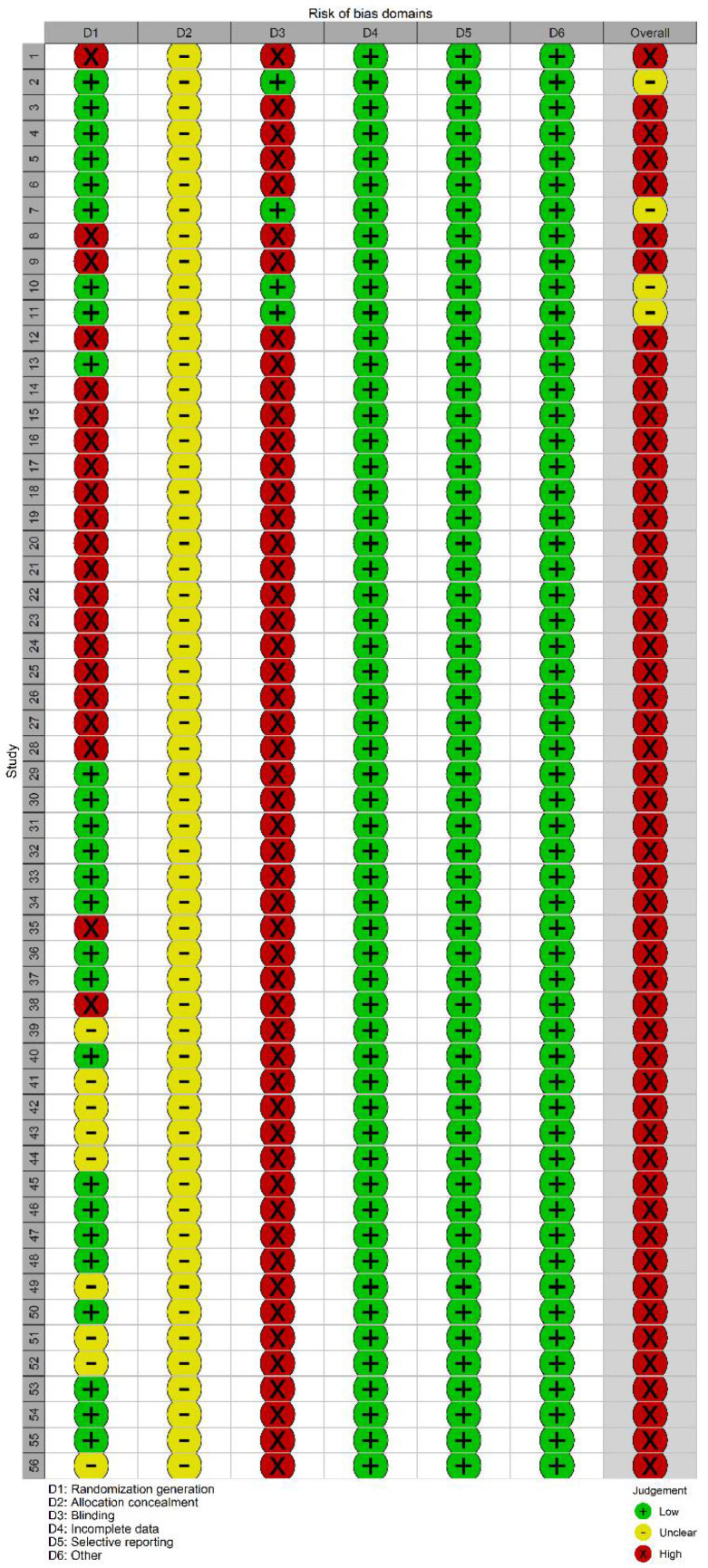
Risk of bias summary.

**Figure 4 pharmaceuticals-15-00953-f004:**
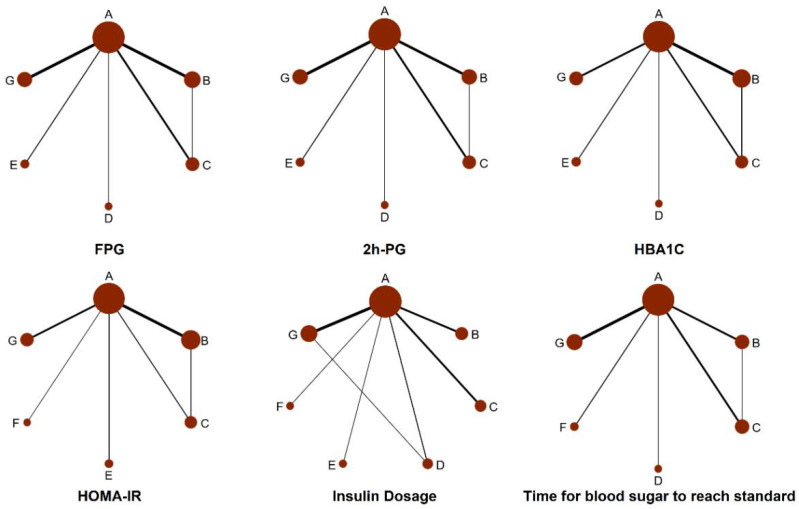
Network plots of different outcomes and direct/indirect comparisons in the network meta-analysis. Figure note: The network diagram shows direct and indirect comparisons between studies. The connection lines represent evidence of direct comparison between the two linked interventions. The two interventions without wires can be compared indirectly using a network meta-analysis. The size of the point represents the sample size of a treatment. The thickness of the line represents the number of studies compared using the two methods. Treatments of the various groups: A: “CSII”, B: “CSII + TZDs”, C: “CSII + Metformin/MF”, D: “CSII + AGIs”, E: “CSII + GLP-1Ras”, F: ”CSII + SGLT2is”, G: “CSII + DPP-4is”.

**Figure 5 pharmaceuticals-15-00953-f005:**
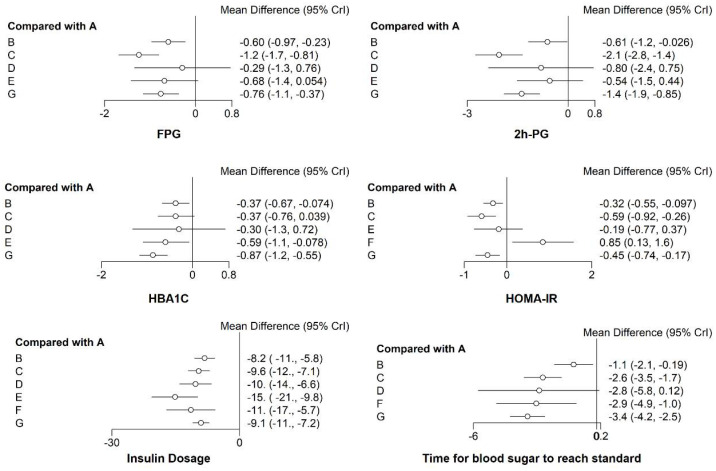
Forest map meta-analysis of different interventions and outcome indicators. Treatments of the various groups: A: “CSII”, B: “CSII + TZDs”, C: “CSII + Metformin/MF”, D: “CSII + AGIs”, E: “CSII + GLP-1Ras”, F: ”CSII + SGLT2is”, and G: “CSII + DPP-4is”.

**Figure 6 pharmaceuticals-15-00953-f006:**
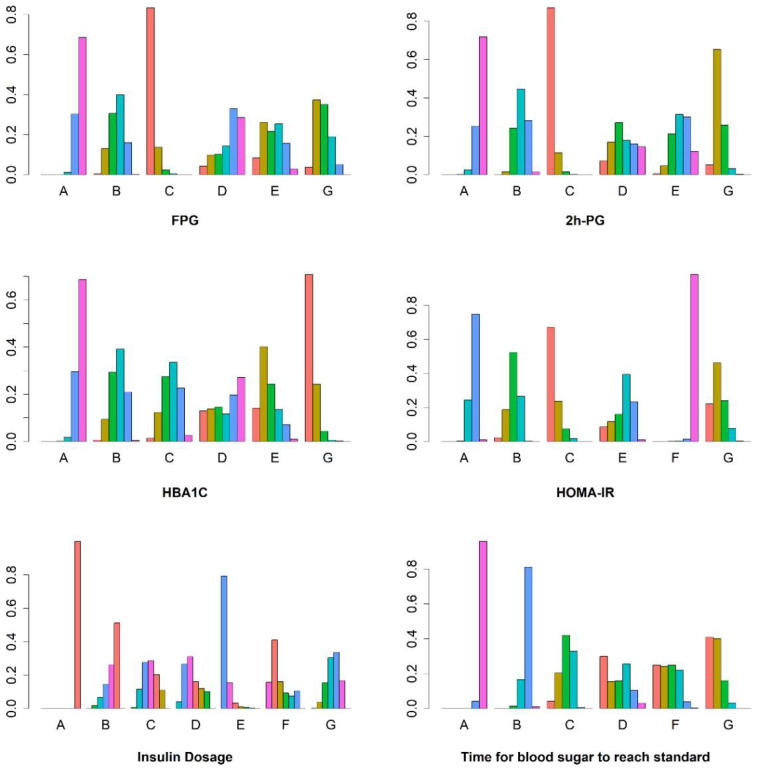
Ranking chart of meta-analyses of different interventions and outcomes. Treatments of the various groups: A: “CSII”, B: “CSII + TZDs”, C: “CSII + Metformin/MF”, D: “CSII + AGIs”, E: “CSII + GLP-1Ras”, F: “CSII + SGLT2is”, and G: “CSII + DPP-4is”.

**Table 1 pharmaceuticals-15-00953-t001:** SUCRA values of different interventions and outcomes.

	A (CSII)	B (CSII + TZDs)	C (CSII + MF)	D (CSII + AGIs)	E (CSII + GLP-1Ras))	F (CSII + SGLT2is)	G (CSII + DPP-4is)
FPG	0.06	0.48	0.96	0.31	0.56	/	0.63
2 h PG	0.06	0.39	0.97	0.48	0.35	/	0.74
HbA1C	0.07	0.46	0.46	0.41	0.68	/	0.93
HOMA-IR	0.25	0.59	0.91	/	0.48	0.01	0.76
Insulin usage	0.00	0.30	0.52	0.61	0.95	0.70	0.43
Blood sugar Standards time	0.01	0.24	0.59	0.64	/	0.69	0.84

## Data Availability

Data sharing not applicable.

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
