# Peer review of "Continuous Subcutaneous Insulin Infusion (CSII) Combined with Oral Glucose-Lowering Drugs in Type 2 Diabetes: A Systematic Review and Network Meta-Analysis of Randomized, Controlled Trials"

_pharmaceuticals, 2022, doi:10.3390/ph15080953_

Round 1

Reviewer 1 Report

Thanks to the authors of the work for the effort made. 

Minor modifications:

1. Restructuring of the work: Material & methods (point 2); Results (point 3); Discussion (point 4).

2. Line 66 needs correction: "alsocorrect"

    Line 84: please, specify what the authors mean "xx studies"

3. Discussion: It needs to add some references, i e, in lines 244 ("...than insulin infusion therapy (reference)") and 272 ("...dosage of insulin (reference).

Author Response

Dear Reviewer,

Thank you for your suggestions for revisions to this article. All the authors have seriously discussed about all these comments. According to your comment, we have tried best modify our manuscript to meet your requirements. In this revised version, changes to our manuscript within the document were all highlighted by using red colored text. Point-by-point responses are listed below this letter. Your kind consideration is highly appreciated.

Yours sincerely, Kai Wang and Li Ma

Corresponding authors

Response to Reviewer 1 Comments

Point 1: Restructuring of the work: Material & methods (point 2); Results (point 3); Discussion (point 4).

Response 1: Thank you for the concern. The structure of the work has been revised, modified as follows: Material & methods (point 2); Results (point 3); Discussion (point 4).

Point 2: â‘ Line 66 needs correction: "alsocorrect". â‘¡Line 84: please, specify what the authors mean "xx studies"

Response 2: Thank you for your reminding. â‘ We modify "alsocorrect" to "also correct". â‘¡We have revised the original text by removing "some concerns in xx studies," and changing "53" to "52".

Point 3: Discussion: It needs to add some references, i e, in lines 244 ("...than insulin infusion therapy (reference)") and 272 ("...dosage of insulin (reference).

Response 3: Thank you for your reminding. We have added the references in the above two places.

Reviewer 2 Report

The manuscript 1786209 entitled " Continuous subcutaneous insulin infusion (CSII) combined with oral glucose-lowering drugs in type 2 diabetes: A systematic review and network meta-analysis of randomized controlled trials" is a review and meta-analysis exploring the best combination of oral antidiabetic drugs and continuous insulin infusion. Authors here describe the best combinations to achieve better therapeutic results. I believe the manuscript has considerable clinical relevance. Due to such relevance, I believe that the main problem with the manuscript is the poor characterization of the patients involved. I believe the authors should include a better characterization of the populations enrolled in the original studies, which will clarify the clinical relevance of the results and the possible use of such results in different populations. I am aware that most of the studies focus on Chinese populations, but this issue should be better addressed.

Author Response

Dear Reviewer,

 Thank you for your suggestions for revisions to this article. All the authors have seriously discussed about all these comments. According to your comment, we have tried best modify our manuscript to meet your requirements. In this revised version, changes to our manuscript within the document were all highlighted by using red colored text. Point-by-point responses are listed below this letter. Your kind consideration is highly appreciated.

Yours sincerely, Kai Wang and Li Ma

Corresponding authors

Response to Reviewer 2 Comments

Point : The manuscript 1786209 entitled " Continuous subcutaneous insulin infusion (CSII) combined with oral glucose-lowering drugs in type 2 diabetes: A systematic review and network meta-analysis of randomized controlled trials" is a review and meta-analysis exploring the best combination of oral antidiabetic drugs and continuous insulin infusion. Authors here describe the best combinations to achieve better therapeutic results. I believe the manuscript has considerable clinical relevance. Due to such relevance, I believe that the main problem with the manuscript is the poor characterization of the patients involved. I believe the authors should include a better characterization of the populations enrolled in the original studies, which will clarify the clinical relevance of the results and the possible use of such results in different populations. I am aware that most of the studies focus on Chinese populations, but this issue should be better addressed.

Response: Thank you very much for your valuable comments and suggestions on our manuscript. The characterization of the patients involved can be seen in the Supplementary Materials: Table S1: PRISMA Checklist. Table S2: Basic characteristics of the studies included. Table S3: Quality evaluation of the studies included. In our research, the number of Chinese studies obtained through strict retrieval strategy is high, while the number of international studies is relatively low. However, the literature we have included is all high-quality literature tested by PRISMA Checklist. We will also further incorporate more ,high-quality RCTs to provide evidence-based support for the use of an insulin pump combined with oral drugs in the treatment of type 2 diabetes.

Reviewer 3 Report

This is an interesting meta analysis on the effects of addition of antidiabetic drugs other than insulin to CSII treatment. The results are consistent with the previous knowledge on the use of metformin and other drugs on diabetes control. The statistical techniques used are beyond my abilities to make a critical analysis. However, I would like to make the following comments, as a clinician:

-There is no data on weight changes induced by added drugs.

-I guess this study concentrated on patients with T2DM, alghough some studies have been conducted in patients with T1DM. Please state it precisely. 

The values are given in the text without units (eg, mmol/L for blood sugar. Please add them.

Author Response

Dear Reviewer,

Thank you for your suggestions for revisions to this article. All the authors have seriously discussed about all these comments. According to your comment, we have tried best modify our manuscript to meet your requirements. In this revised version, changes to our manuscript within the document were all highlighted by using red colored text. Point-by-point responses are listed below this letter. Your kind consideration is highly appreciated.

Yours sincerely, Kai Wang and Li Ma

Corresponding authors

Response to Reviewer 3 Comments

Point 1:There is no data on weight changes induced by added drugs.

Response 1: Thank you for the concern. Since most of the raw data didn't have data on weight, so there is no data on weight changes induced by added drugs. This is also the weakness of this manuscript. Indeed, change in weight is a clinical indicator of great concern for clinicians in treating diabetes. In future studies, we will perform this analysis by further expanding the sample size to obtain the effect of insulin pumps in combination with different oral drugs on body weight.

Point 2: I guess this study concentrated on patients with T2DM, alghough some studies have been conducted in patients with T1DM. Please state it precisely.

Response 2: Dear the reviewers, this study entirely on patients with T2DM.

Point 3:The values are given in the text without units (eg, mmol/L for blood sugar. Please add them.

Response 3: Thank you for your reminding. We have carefully examined the units problems mentioned above, and made all modifications.

Round 2

Reviewer 2 Report

The authors have improved the manuscript quality according to previous comments

This manuscript is a resubmission of an earlier submission. The following is a list of the peer review reports and author responses from that submission.